# A Comparative Systematic Review of COVID-19 and Influenza

**DOI:** 10.3390/v13030452

**Published:** 2021-03-10

**Authors:** Molka Osman, Timothée Klopfenstein, Nabil Belfeki, Vincent Gendrin, Souheil Zayet

**Affiliations:** 1Faculty of Medicine of Tunis, University Tunis El Manar, Tunis 1007, Tunisia; molkaosman@gmail.com; 2Infectious Disease Department, Nord Franche-Comté Hospital, 90400 Trévenans, France; timothee.klopfenstein@hnfc.fr; 3Internal Medicine Department, Groupe Hospitalier Sud Ile de France, 77000 Melun, France; nabil.belfeki@ghsif.fr

**Keywords:** influenza, COVID-19, SARS-CoV-2, clinical features, laboratory, imaging, systematic review

## Abstract

Background: Both SARS-CoV-2 and influenza virus share similarities such as clinical features and outcome, laboratory, and radiological findings. Methods: Literature search was done using PubMed to find MEDLINE indexed articles relevant to this study. As of 25 November 2020, the search has been conducted by combining the MeSH words “COVID-19” and “Influenza”. Results: Eighteen articles were finally selected in adult patients. Comorbidities such as cardiovascular diseases, diabetes, and obesity were significantly higher in COVID-19 patients, while pulmonary diseases and immunocompromised conditions were significantly more common in influenza patients. The incidence rates of fever, vomiting, ocular and otorhinolaryngological symptoms were found to be significantly higher in influenza patients when compared with COVID-19 patients. However, neurologic symptoms and diarrhea were statistically more frequent in COVID-19 patients. The level of white cell count and procalcitonin was significantly higher in influenza patients, whereas thrombopenia and elevated transaminases were significantly more common in COVID-19 patients. Ground-grass opacities, interlobular septal thickening, and a peripheral distribution were more common in COVID-19 patients than in influenza patients where consolidations and linear opacities were described instead. COVID-19 patients were significantly more often transferred to intensive care unit with a higher rate of mortality. Conclusions: This study estimated differences of COVID-19 and influenza patients which can help clinicians during the co-circulation of the two viruses.

## 1. Introduction

Coronavirus disease 2019 (COVID-19), an infection caused by severe acute respiratory syndrome coronavirus-2 (SARS-CoV-2), was first reported in December 2019, in Wuhan, province of Hubei in China. Since then, it has evolved into a worldwide pandemic and became one of the world’s toughest health problems [1]. Both SARS-CoV-2 and influenza virus share similarities such as viral shedding, route of transmission, and clinical presentation [2]. Influenza affects 5–10% of adults annually with most cases occurring during the winter season [3]. According to the World Health Organization (WHO), influenza’s epidemics are estimated to result in about 3 to 5 million cases of severe illness, and about 290,000 to 650,000 respiratory deaths [4].

There have been few studies comparing COVID-19 with influenza mainly focused on epidemiological features in the general population [5]. However, no review study compared clinical, biological, and radiological characteristics between COVID-19 and influenza. With projections estimating the COVID-19 pandemic to last for another year [2] and it being the influenza season, it is crucial to evaluate the differences between the two diseases. The current study performed a systematic review to compare the clinical features and outcome, laboratory, and radiological findings of COVID-19 patients with influenza adult patients.

## 2. Methods

This study is based on a review of published literature comparing COVID-19 and influenza. Literature search was done using PubMed to find MEDLINE indexed articles relevant to this study. As of 25 November 2020, the search has been conducted by combining the MeSH words “COVID-19” and “Influenza”. At first, citations that were not in English or in French and not a free full text were excluded. Then, we screened citations based on titles and abstracts. Those with irrelevant information or subjects, studies with focus other than comparing COVID-19 and influenza, studies with a population under 18 years old and case reports about coinfection between COVID-19 and influenza were excluded. A *p*-value < 0.05 was considered significant to compare the two viruses.

## 3. Results

### 3.1. Included Citations in the COVID-19 vs. Influenza Review

On PubMed, 331 articles were initially found, out of which 46 articles were selected. In the end, 18 articles were selected [3,5,6,7,8,9,10,11,12,13,14,15,16,17,18,19,20,21] (8 retrospective studies, 6 cohort studies, 3 case-control studies, 1 prospective study, 1 cross-sectional study, 1 review and meta-analysis, and 1 review) (Figure 1 and Table 1)

### 3.2. Studies in Patients with COVID-19 and Influenza

#### 3.2.1. Demographic and Clinical Findings

In this review, 14 studies of 18 [5,6,9,10,11,12,13,14,15,16,18,19,20,21] reported a comparison between COVID-19 and influenza patients based on demographic findings, comorbidities, clinical features, and outcome (Table 2). Fever and respiratory symptoms such as cough, expectoration or sputum production and dyspnea were the main symptoms in both groups with COVID-19 and influenza; however, they were significantly more frequent in patients with influenza [5,9,10,11,13,15,16,19,20]. Moreover, vomiting, otorhinolaryngological symptoms such as nasal congestion, rhinorrhea, sore throat and ocular symptoms such as tearing and conjunctival hyperhemia were statistically more frequent in patients with influenza than COVID-19 adult patients [5,10,11,12,15]. In patients infected with COVID-19, the most significant and frequent clinical features were fatigue, neurologic symptoms such as headache (especially facial headache: retro-orbital or frontal headache), anosmia and dysgeusia, gastro-intestinal (GI) symptoms such as diarrhea and acute respiratory distress syndrome (ARDS), compared to influenza patients [11,13,14,15,18]. In a retrospective case-control study, Tang et al. reported that patients with influenza were more disposed to have productive cough and higher Sequential Organ Failure Assessment (SOFA) scores (OR = 9.58, (95% CI = 1.73–64.72), *p* = 0.011 and OR = 2.26 (1.12–3.57), *p* = 0.006, respectively). Meanwhile, COVID-19 patients exhibit symptoms of fatigue or GI symptoms (OR = 0.91 (0.84–0.98), *p* = 0.011; OR = 0.12 (0.02–0.94), *p* = 0.013 and OR = 0.10, (0.01–0.98), *p* = 0.044, respectively) [13].

Concerning comorbidities; cardiovascular disease/hypertension, diabetes and obesity were significantly higher in patients with COVID-19, while respiratory diseases such as asthma and chronic obstructive pulmonary disease (COPD) and immunocompromised conditions were significantly more common in influenza patients [5,11,18]. Zayet et al. concluded that COVID-19 patients had a lower Charlson comorbidity index (CCI) than influenza patients (2 ± 2.5 vs. 3 ± 2.4, respectively, *p* = 0.003), but with no significant differences regarding comorbidities [15]. Qu et al. reported that while the incidences of COVID-19 and influenza were comparable among the 18–65 and >65 year age groups, the incidences of influenza were much higher than COVID-19 among those aged under 18 years old [9]. However, Lee et al., reported that the median age was significantly higher in patients with COVID-19 compared to patients with influenza (68 (IQR: 59–75) years vs. 57 (IQR: 44–63) years, *p* = 0.016) [21]. No significant difference was found concerning other demographic characteristics such as gender and ethnicity in our review.

Only a few studies (2/18 studies in adults) were interested in describing the timeline and onset of the main symptoms in COVID-19 and influenza [14,15]. Zayet et al. reconstituted the natural history of the two viruses and suggested that the onset of symptoms (pain symptoms appeared first, followed by fever, cough and diarrhea) did not differ between the two groups except fever, which appeared earlier in COVID-19 than in influenza (1.9 days vs. 2.5 days, *p* = 0.045) [15]. They also concluded that hospitalization and clinical aggravation occurred later in COVID-19 than in influenza. In fact, COVID-19 patients were hospitalized at day 7 ± 3 (vs. day 5 ± 2 in influenza, *p* = 0.038), had a respiratory rate ≥ 22/min at day 9 ± 0.8 (vs. day 5 ± 1.3 in influenza, *p* < 0.001), and were admitted to intensive care unit (ICU) at day 10 ± 2.7 (vs. day 7 ± 2.4 in influenza, *p* < 0.004). Cobb et al. also reported that patients with COVID-19 had a longer median symptom duration prior to hospitalization compared to patients with influenza (7 days (IQR 5–13) vs. 3.5 (IQR 2–7), *p* < 0.001) [14]. In the prospective study of Raija Auvinen et al., in multivariable Cox regression analysis, the predictors associated with a longer duration of hospitalization were COVID-19 (hazard ratio (HR) 0.221, (95% CI = 0.118–0.416), *p* < 0.001), age (HR = 0.972, (0.955–0.990), *p* = 0.002), Body Mass Index (HR = 0.950 (0.916–0.986), *p* = 0.006) and diabetes (HR 0.539, (0.272–1.066), *p* = 0.076) [18].

#### 3.2.2. Laboratory Findings

A total of 11 studies of 18 [7,8,9,10,11,13,14,18,19,20,21] compared routine laboratory test results between COVID-19 and influenza patients. Both infections caused abnormalities in biological parameters leading to aberrant blood cell counts and sometimes elevated hepatic, renal, and cardiac enzyme activity. White blood cells, especially neutrophils (NE) were significantly more elevated in influenza than COVID-19 patients [7,8,10,11,14,18,19,20,21]. Lymphocytes proportion in white blood cell count are higher in influenza than COVID-19 patients [7,8,20] but without difference between lymphocytes level [7,8], except for one study [20]. Platelets level was lower [11] with more often thrombocytopenia [18] in COVID-19 than influenza patients. Elevated transaminases were significantly more common in COVID-19 patients [7,8,11,18,20]. In a prospective study by Raija Auvinen et al, it was reported that C-reactive protein (CRP) values were similar at admission but rose significantly higher in COVID-19 patients than influenza patients during hospitalization [18]. However, elevated procalcitonin was significantly more frequent in influenza patients than in COVID-19 patients [19,20]. COVID-19 patients had also a greater disposition to elevated prothrombin time (OR = 0.63, 95% CI (0.46–0.86), *p* = 0.004) [13].

Finally, Chen et al. developed a diagnostic formula to differentiate between COVID-19 and influenza during their early stages. Two parameters (monocyte (MO) count and percentage of basophils (BA)) were combined to clarify the diagnostic efficacy, with a sensitivity of 71.6% and a specificity of 74.8% (joint probability (*P*) = 2.388 × BA% − 5.182 × MO# + 2.192 (AUC 0.772, 95% CI (0.718–0.826)). COVID-19 should be considered as the diagnosis when the joint probability is greater than 0.45, while influenza should be considered when it is less than 0.45 [8] (Table 3).

#### 3.2.3. Radiological Findings

We found 10 studies of 18 on adult patients [10,11,12,13,14,16,17,18,19,20] that described imaging findings in COVID-19 and influenza, especially in chest computed tomography (CT) manifestations. Ground-grass opacities (GGO), interlobular septal thickening, and a peripheral distribution were more common in patients with COVID-19 than in patients with influenza [10,12,13,16,18,20]. However, consolidation, nodules, and linear opacities were more common in patients with influenza than those with COVID-19 [11,12,13,18,19]. Tang et al. reported that influenza patients were more inclined to have consolidation manifested on chest CT imaging (OR = 4.95 (95% CI = 1.518–16.176), *p* = 0.008), whereas COVID-19 patients had a greater disposition for having GGO on chest CT scans (OR = 0.086 (0.015–0.490); *p* = 0.006) [13]. In a review from Onigbinde et al., GGO were usually peripherally located and that vascular engorgement, pleural thickening, and subpleural lines were more frequent in COVID-19 patients. Pneumomediastinum and pneumothorax were only reported in influenza studies [3] (Table 4).

## 4. Discussion

The current review compared the clinical features and courses, laboratory data, and radiological findings between COVID-19 patients and those with influenza. We initially found that compared to patients with influenza, COVID-19 patients were more likely to exhibit symptoms such as diarrhea and neurologic symptoms. However, fever, productive cough, vomiting, dyspnea, ocular and otorhinolaryngological symptoms were more observed in the group influenza than the group COVID-19. Some authors concluded by describing the timeline of these viruses that SARS-CoV-2 infection may present with a slow onset compared with the clinical presentations of influenza infection with a longer incubation period [13,15]. Indeed, the differences in the clinical presentation of these two viral infections can be explained, in large part, by the pathophysiological distribution of the entry receptors for each virus. Human influenza A virus binds to cell receptors alpha2,6-linked via sialic acid linked glycoproteins. The distribution of sialic acid on cell surfaces is one determinant of host tropism and understanding its expression on human cells and tissues is important for understanding influenza pathogenesis. These receptors were especially expressed on the respiratory tract, from the nasopharynx, trachea to the bronchi, except the alveoli (α2,3-linked sialic acid receptors predominant on alveolar cells) [22]. The short incubation period of influenza infection with the predominant respiratory manifestations (sore throat, sneezing, sputum production, rhonchi on pulmonary auscultation) is well explained by this distribution. On the other hand, Angiotensin-converting enzyme 2 (ACE2) protein, known as the key regulator enzyme of the renin–angiotensin–aldosterone system (RAAS) is the functional receptor of SARS-CoV-2 and its expression and activity will mediate directly the SARS-CoV-2 infection. Regarding the tissue distribution of ACE2 protein, ACE2 is highly expressed on lung alveolar epithelial cells, small intestinal epithelial cells and endothelial cells (including in the central nervous system), but poorly found on the surface of nasopharyngeal cells [23]. When the SARS spike protein binds to the ACE-2 receptor, the complex is proteolytically processed by type 2 transmembrane protease TMPRSS2 leading to cleavage of ACE-2 and activation of the spike [24]. This mechanism is also described in influenza physiopathology. All these findings trigger a longer incubation period of SARS-CoV-2 infection and the observed symptoms in COVID-19 patients of dyspnea, dry cough, diarrhea, and bilateral crackling sound on pulmonary auscultation, but also neurologic symptoms such as new loss of smell and taste. ACE is also abundantly present in the basal layer of the non-keratinizing squamous epithelium of nasal and oral mucosa [25]. Indeed, most reports have so far linked new loss of taste or smell to neurological symptoms instead of rhinolaryngological symptoms [26]. In any case, in this epidemic context, patients presenting with dysgeusia and/or anosmia may be considered as patients infected with COVID-19, until microbiological confirmation has been obtained. In another study including 217 patients presenting influenza-like illness, we demonstrated that the specificity of the combination of anosmia and dysgeusia reached 91% for a positive SARS-CoV-2 RT-PCR result [27]. Unspecific symptoms such as fever and musculoskeletal symptoms or pain syndrome defined by fatigue, myalgia and/or arthralgia are associated with a cascade of inflammatory mediators and were not directly linked to the distribution of viral receptors. These clinical presentations can be equally described in the two illnesses. Concerning the outcome, both COVID-19 and influenza may be accompanied by ARDS with a high mortality [13]. However, it was clear that mortality and lethality of SARS-CoV-2 infection were much higher than those of the influenza infection in the general population [28].

Comparison of peripheral blood parameters from COVID-19 and influenza patients revealed some differences, especially during the early stages of the disease. Concerning the blood count, NE were significantly more elevated in influenza than COVID-19 patients. Jiangnan Chen et al. conducted their study with a group control [8], the difference about NE is explained by an increase of NE in influenza and a decrease of NE in SARS-CoV-2 infection. The only study which has shown a higher level of lymphocytes in influenza than SARS-CoV-2 infection [20] is not conclusive because the two populations were not comparable (18% received invasive mechanical ventilation (IMV) in the influenza group while none received IMV in the COVID-19 group). Influenza and SARS-CoV-2 infection seems to decrease lymphocytes level with the same intensity. In another short review of COVID-19 hematological manifestations, Slomka et al. showed that the majority of patients (especially those presenting with an ARDS related to SARS-CoV-2) are likely to develop lymphocytopenia. The decreased number of lymphocytes is described in other diseases caused by other coronaviruses primarily infecting the human respiratory tract [29]. This lymphopenia is probably explained by direct infection of lymphocytes and destruction of lymphoid organs caused by the “cytokine storm” and a major release of pro-inflammatory cytokines [30]. On the one hand, it is known that MO and macrophages play central roles in the immune response of humans and in protecting the body from influenza infection. They are necessary for the influenza virus to infect lymphocytes and regulate lymphocyte apoptosis by synthesizing and expressing viral neuraminidase [31]. On the other hand, several studies revealed that COVID-19 patients exhibited a significant decrease under lower limit of lymphocyte counts [32]. This indicates that SARS-CoV-2 consumes many immune cells and inhibits the body’s cellular immune function [33]. Conversely, inflammatory cytokines including serum amyloid A (SAA), CRP, IL-6, IL-10 were significantly higher than normal values [32]. SARS-COV-2 invasion activated T cell-mediated immunity, which resulted in increasing production of inflammatory cytokine [34]. However, inhibition cytokines such as IL-10 can suppress T cell activation conversely. Serum amyloid A and CRP are both of acute-phase proteins in response to inflammatory cytokines related by activated MO/macrophages after infections. High levels of SAA and CRP could reflect the severity of inflammation and result in a “cytokine storm”, during which the inflammation spreads throughout the body via the circulation. This severe inflammatory state secondary to COVID-19 leads to a severe derangement of hemostasis that has been recently described as a state of hypercoagulopathy, defined as increased degradation products such as D-dimer and fibrinogen. In a large American cohort study of 9407 adult COVID-19 patients, the overall in-hospital venous thromboembolism (VTE) was approximately 3%. The authors concluded that key predictors of VTE or mortality included advanced age, increasing CCI, past history of cardiovascular disease, ICU level of care, and elevated level of D-dimer [35]. In addition, several radiologists tried to differentiate COVID-19 from Influenza and other viral pneumonia on chest CT images [36]. Usually, the common findings on CT for typical influenza pneumonia consist of diffuse or multifocal GGO and small centrilobular nodules [3]. Onigbinde et al. showed that the occurrence of GGO and consolidation in COVID-19 were not markedly different from that of influenza studies. However, they were predominantly located bilaterally within the lower lobes for COVID-19 patients, whereas in the influenza studies, they were more widespread, involving all the lobes [3]. In another study including 122 patients, Mengqi et al. concluded that the COVID-19 patients were more likely to have rounded opacities and interlobular septal thickening compared with the influenza group, but less likely to have nodules, tree-in-bud sign and pleural effusion with significant difference between the two groups [12]. In our review, complications such as pneumothorax and pneumomediastinum were only described in influenza patients. These respiratory complications remain scarce during the course of COVID-19 and have recently been described in medical literature [3,37,38]. Therefore, these differential pathologic changes may present themselves as distinguishing imaging characteristics during clinical assessments. There are some limitations to our review. First of all, COVID-19 is relatively novel with a limited number of patients and studies. In most articles, the clinical features, laboratory data and CT findings reporting on influenza dated before the onset of COVID 19. Finally, the use of abstracts in selecting full texts for a detailed review could have led to the omission of some articles. Moreover, we only considered PubMed a reference for our bibliographic research and database with accessible free access articles. We also just summarized what we found in recent medical literature and focused on what was statistically significant. Finally, a meta-analysis is recommended to further define the differences and the degree between COVID-19 and influenza.

## 5. Conclusions

This study provided a comprehensive comparison of adult patients in SARS-CoV-2 and influenza infections, regarding comorbidities, clinical and paraclinical features, and outcome. Clinical manifestations of COVID-19 and influenza seem to be similar with some differences: neurologic symptoms and diarrhea were more described in COVID-19, however; vomiting, ocular and otorhinolaryngological symptoms were more observed in influenza infection. Both viruses decreased lymphocytes; NE were significantly more elevated in influenza than COVID-19 patients while elevated transaminases were significantly more elevated in COVID-19 than influenza patients. Radiological findings showed that GGO are usually peripherally located in COVID-19 compared with influenza which also had central and random locations. All these findings can help clinicians when dealing with cases of influenza-like illnesses during the period of co-circulation of influenza and SARS-CoV-2.

## Figures and Tables

**Figure 1 viruses-13-00452-f001:**
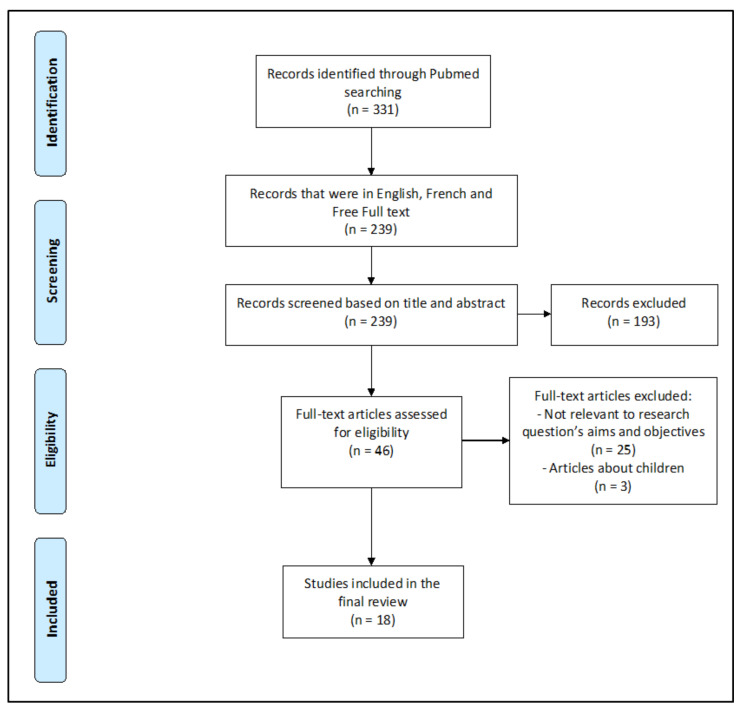
PRISMA guideline flowchart detailing article selection process.

**Table 1 viruses-13-00452-t001:** General summary of included citations.

References	Country	Study Type **	Number of Patients/Studies * (COVID-19 vs. Influenza)
Jordan Cates et al. [6]	United States	Cohort	9401 (3948 vs. 5453)
Ying Luo et al. [7]	China (Hubei)	Cohort	2167 (1027 vs. 1140)
Jiangnan Chen et al. [8]	China (Shaoxing)	Case-Control	380 (169 vs. 131; 80 healthy controls)
Jiajia Qu et al. [9]	China	Retrospective Cohort	366 (246 vs. 120)
Jianguo Zhang et al. [10]	China	Retrospective cohort	326 (211 vs. 115)
Helene Faury et al. [11]	France (Paris)	Retrospective	200 (100 vs. 100)
Pengfei Li et al. * [5]	-	Systematic review and Meta-analysis	197 (113 vs. 84)
Mengqi Liu et al. [12]	China (Chongqing)	Retrospective	180 (122 vs. 48)
Xiao Tang et al. [13]	China (Wuhan)	Retrospective case-control	148 (73 vs. 75)
Natalie L. Cobb et al. [14]	United States (Washington)	Retrospective Cohort	139 (65 vs. 74)
Souheil Zayet et al. [15]	France	Retrospective	124 (70 vs. 54)
Hao Wang et al. [16]	China	Retrospective	105 (13 vs. 92)
Liaoyi Lin et al. [17]	China (Wenzhou)	Retrospective	97 (52 vs. 45)
Raija Auvinen et al. [18]	Finland	Prospective study	61 (28 vs. 33)
Zhilan Yin et al. [19]	China	Retrospective	60 (30 vs. 30)
Yi-Hua Lin et al. [20]	China (Xiamen)	A cross-sectional retrospective study	57 (35 vs. 22)
Jaehee Lee et al. [21]	South Korea (Daegu)	Retrospective	29 (20 vs. 09)
Stephen O. Onigbinde et al. * [3]	-	Review	17 (09 vs. 08)

* Studies for Review or Meta-analysis. ** Cohort studies are used to investigate causes of disease and establish association between risk factors and health outcomes. An outcome-free study population is first identified by the exposure/event of interest and followed in time until the outcome of interest happens. They can be prospective (carried out from the present time into the future) or retrospective (carried out at the present time and look to the past to examine medical events or outcome). Case-Control studies first identify subjects by outcome status (cases), then select from the same source population, subjects without the outcome (control). Cross-sectional study or prevalence study is an observational study that collects data on the subjects of interest at a specific point in time.

**Table 2 viruses-13-00452-t002:** Significant demographic and baseline characteristics, clinical features, and outcome in COVID-19 and influenza adult groups.

References	Significant Clinical Features/Outcome	COVID-19 (%)	Influenza (%)	*p*-Value < 0.05
Jordan Cates et al. [6]	Admitted to ICU	36.5	17.6	<0.001
	Hospital mortality	21	3.8	<0.001
	Duration of hospitalization (median days, [IQR])	**8.6 [3.9–18.6]**	3.0 [1.8–6.5]	<0.001
Jiajia Qu et al. [9]	Fever	78.5	**89.2**	<0.05
	Persistent fever	50.4	**74.2**	<0.01
Jianguo Zhang et al. [10]	Cough	69.7	**86.1**	0.001
	Expectoration	22.7	**74.8**	<0.001
	Dyspnea	14.7	**27.8**	0.004
	Chest pain	13.7	**27.8**	0.002
	Vomiting	1.4	**9.6**	<0.001
Helene Faury et al. [11]	Chronic pulmonary diseases	12.0	**27.0**	0.01
	Overweight/Obesity	**40.8**	25.0	0.02
	Median BMI	**27.3**	24.8	0.04
	Fatigue	**63.6**	39.0	0.0006
	Diarrhea	**25.8**	13.0	0.03
	Faintness	**12.1**	3.0	0.02
	Anosmia/Ageusia	**7.0**	0	0.01
	Sputum production	12.0	**36.0**	0.0001
	Nasal Congestion	8.3	**21**	0.02
	Secondary respiratory failure	**21.0**	0	<0.0001
	Acute Kidney failure	**17.0**	7.0	0.048
	Pulmonary embolism	**6.0**	0	0.03
	Heart congestion	2.0	**14.0**	0.003
	Admitted to ICU	**31.0**	12.0	0.002
	Duration of hospitalization (days, [IQR])	**10 [4–17]**	4 [1–11]	<0.0001
	Oxygen therapy	**65.0**	42.3	0.002
	Mortality rate	**20.0**	5.0	0.002
Pengfei Li et al. [5]	Cardiovascular disease/Hypertension	**28.76**	14.11	<0.0001
	Diabetes	**16.38**	11.12	0.012
	Asthma	8.42	**16.09**	0.0033
	Chronic Obstructive Pulmonary disease	4.93	**9.52**	0.0003
	Immunocompromised conditions	4.39	**9.99**	<0.0001
	Fever	72.08	**89.99**	<0.0001
	Cough	57.99	**85.31**	<0.0001
	Shortness of breath	32.89	**49.19**	0.0249
	Rhinorrhea	8.48	**38.57**	<0.0001
	Sore throat	9.48	**37.28**	<0.0001
	Myalgia/Muscle pain	18.97	**30.12**	0.0242
	Vomiting	8.67	**24.27**	<0.0001
Mengqi Liu et al. [12]	Stuffy and runny nose	7	**23**	0.002
Xiao Tang et al. [13]	Productive cough	53.4	**78.7**	0.002
	Fatigue	**63**	18.7	<0.001
	GI symptoms	**37**	6.7	<0.001
	Myalgia	**34.2**	14.7	0.007
Natalie L. Cobb et al. [14]	ARDS	**63**	26	<0.001
	Hospital mortality	**40**	19	0.006
Souheil Zayet et al. [15]	Frontal headache	**25.7**	9.3	0.021
	Retro-orbital or temporal headache	**18.6**	3.7	0.013
	Fever	75.7	**92.6**	0.042
	Anosmia	**52.9**	16.7	<0.001
	Dysgeusia	**48.6**	20.4	0.001
	Diarrhea	**40**	20.4	0.021
	Sputum Production	28.6	**51.9**	0.01
	Sneezing	18.6	**46.3**	0.001
	Dyspnea	34.3	**59.3**	0.007
	Sore throat	20	**44.4**	0.006
	Conjunctival hyperemia	4.3	**29.6**	<0.001
	Tearing	5.7	**24.1**	0.004
	Vomiting	2.8	**22.2**	0.001
	Crackling sound	**38.6**	20.4	0.032
	Ronchi sounds	1.4	**16.7**	0.002
Hao Wang et al. [16]	Cough	30.8	**82.6**	0
Raija Auvinen et al. [18]	Pulmonary Diseases	18	**45**	0.03
	Current smoking	4	**30**	0.008
	Headache	**85**	52	0.004
	ARDS	**93**	58	0.003
	ICU admission	**29**	6	0.034
	Duration of hospitalization (days, [IQR])	**6 [4–21]**	3 [2–4]	<0.001
Zhilan Yin et al. [19]	Cough	73.3	**96.7**	0.026
	Expectoration	43.3	**80**	0.007
Yi-Hua Lin et al. [20]	Fever 38.0 °C–38.9 °C	**43**	32	0.014
	Fever ≥39.0 °C	11	**45**	0.014
	Cough	51	**100**	<0.001
	Expectoration	28	**91**	<0.001
	Dyspnea	9	**59**	<0.001
	Chills	23	**55**	0.015
Jaehee Lee et al. [21]	Median heart rate (bpm)	83	**107**	0.017

Bold: illness (COVID-19 or influenza) with significant difference. Abbreviations: ARDS: Acute respiratory distress syndrome; ICU: Intensive care unit; IQR: interquartile range.

**Table 3 viruses-13-00452-t003:** Significant laboratory findings in COVID-19 and influenza adult groups.

References	Significant Laboratory Findings	COVID-19 (%)	Influenza (%)	*p*-Value < 0.05
Ying Luo et al. [7]	White blood cell count (×10^9^ /L, median, [IQR])	5.45 [4.46–7.17]	**6.14 [4.66–8.24]**	<0.001
	Neutrophil (×10^9^ /L, median, [IQR]))	3.68 [2.68–5.16]	**4.09 [2.85–6.11]**	<0.001
	Lymphocyte (%, median, [IQR])	**22.0 [14.6–29.4]**	20.5 [13.3–28.6]	0.009
	Monocyte (×10^9^ /L, median, [IQR])	0.47 [0.34–0.61]	**0.52 [0.37–0.69]**	<0.001
	Eosinophil (×10^9^ /L, median, [IQR])	0.01 [0.00–0.05]	**0.02 [0.00–0.07]**	<0.001
	Eosinophil (%, median, [IQR])	0.2 [0.0–2.9]	**0.3 [0.0–1.2]**	<0.001
	Basophil (%, median, [IQR])	0.2 [0.0–0.3]	**0.2 [0.1–0.3]**	<0.001
	Red blood cell count (×10^12^ /L, median, [IQR])	**4.43 [4.00–4.84]**	4.37 [3.96–4.78]	0.012
	Hemoglobin (g/L, median, [IQR])	**134 [122–146]**	131 [119–143]	<0.001
	Hematocrit (%, median, [IQR])	**39.7 [36.2–43.1]**	39.1 [35.5–42.4]	0.002
	MCV (fL, median, [IQR])	89.1 [86.4–91.7]	**89.6 [86.7–92.4]**	0.003
	MCH (pg, median, [IQR])	**30.6 [29.5–31.6]**	30.4 [29.3–31.3]	0.002
	MCHC (g/L, median, [IQR])	**343 [335–351]**	337 [329–346]	<0.001
	RDW-CV (Median, [IQR])	12.2 [11.9–12.8]	**12.5 [12.0–13.2]**	<0.001
	RDW-SD (fL, median, [IQR])	39.5 [37.8–41.8]	**40.9 [38.8–43.2]**	<0.001
	PDW (fL, median, [IQR])	12.0 [10.8–13.6]	**12.3 [11.0–13.9]**	0.021
	Alanine aminotransferase (U/L, median, [IQR])	**25 [18–38]**	**24 [16–36]**	0.019
	Aspartate aminotransferase (U/L, median, [IQR])	**27 [21–36]**	25 [19–35]	<0.001
	Total Protein (g/L, mean)	**69.3 ± 5.6**	68.5 ± 6.4	0.003
	Globulin (g/L, median, [IQR])	**32.4 ± 4.4**	31.8 ± 4.8	<0.001
	Indirect Bilirubin (μmol/L, median, [IQR])	5.5 [4.2–7.3]	**4.9 [3.8–6.9]**	<0.001
	GGT (U/L, median, [IQR])	30 [21–48]	**35 [21–54]**	0.003
	Alkaline Phosphatase (U/L, median, [IQR])	65 [56–78]	**75 [63–96]**	<0.001
	LDH (U/L, median, [IQR])	**260 [217–327]**	235 [196–298]	<0.001
	Triglyceride (mmol/L, median, [IQR])	**1.75 ± 088**	1.63 ± 0.84	<0.001
	HDL-C (mmol/L, median, [IQR])	**0.99 ± 0.19**	0.97 ± 0.22	0.002
	LDL-C (mmol/L, median, [IQR])	**2.45 ± 0.55**	2.41 ± 0.68	0.004
	Creatinine (μmol/L, median, [IQR])	**72 [61–87]**	69 [59–82]	<0.001
	Urea (mmol/L, median, [IQR])	**5.89 ± 3.84**	5.54 ± 3.41	0.001
	Uric acid (μmol/L, median, [IQR])	253 [206–313]	**260 [219–304]**	0.031
	Calcium (mmol/L, median, [IQR])	2.14 ± 0.11	**2.17 ± 0.11**	<0.001
	Magnesium (mmol/L, median, [IQR])	**0.87 ± 0.07**	0.86 ± 0.09	0.001
	Chlorine (mmol/L, median, [IQR])	100.4 ± 4.2	**101.4 ± 3.7**	<0.001
	Potassium (mmol/L, median, [IQR])	**4.21 ± 0.42**	4.15 ± 0.40	<0.001
	Sodium (mmol/L, median, [IQR])	**139.7 ±3.9**	139.1 ± 3.4	<0.001
	Phosphate (mmol/L, median, [IQR])	1.04 ± 0.26	**1.05 ± 0.20**	0.002
	HCO^3^ (mmol/L, median, [IQR])	**24.5 ± 2.9**	24.0 ± 3.1	<0.001
	Hypersensitive CRP (mg/L, median, [IQR])	**20.0 [5.8–45.8]**	15.7 [4.8–40.1]	0.024
	ESR (mm/h, median, [IQR])	**35 [24–47]**	27 [17–40]	<0.001
	Prothrombin time (s, mean)	14.06 ± 1.09	**14.09 ± 1.83**	<0.001
	APTT (s, mean)	**39.9 ± 4.5**	39.6 ± 5.0	0.02
	Thrombin time (s, mean)	**16.9 ± 1.4**	16.6 ± 2.0	<0.001
	Prothrombin activity (%, mean)	91 ± 11	**92 ± 14**	<0.001
	Fibrinogen (g/L, mean)	**4.71 ± 1.08**	4.27 ± 1.18	<0.001
	D-Dimer (mg/L, median, [IQR])	1.24 [0.65–2.75]	**1.72 [0.85–3.30]**	<0.001
Jiangnan Chen et al. [8]	Monocyte (×10^9^ /L, median, [IQR])	0.36 [0.28–0.48]	**0.55 [0.4–0.71]**	0
	Monocyte (%, median, [IQR])	7.60 [6.20–9.95]	**9.0 [7.20–11.40]**	0
	Neutrophil (×10^9^ /L, median, [IQR])	2.93 [2.26–3.79]	**4.26 [3.00–5.74]**	0
	Neutrophil (%, mean)	64.50 ± 11.64	**68.42 ± 14.69**	0.011
	Lymphocyte (%, mean)	**26.30 ± 10.52**	21.07 ± 12.85	0
	Eosinophil (%, median, [IQR])	**0.60 [0.30–1.15]**	0.40 [0.10–1.10]	0.038
	Basophil (%, median, [IQR])	**0.20 [0.10–0.30]**	0.10 [0.10–0.30]	0.001
Jiajia Qu et al. [9]	Elevated lymphocyte	0	**5**	<0.01
	Abnormal Urinary test	**32.11**	21.67	<0.05
	Urine protein positive	**16.26**	8.33	<0.05
	Elevated procalcitonin	**40.83**	10.98	<0.01
	Elevated white blood cells	**75**	26.83	<0.01
Jianguo Zhang et al. [10]	Leukocytosis > 9.5 × 10^9^ /L	16.1	**30.4**	0.003
	Neutrophilia > 75%	32.2	**50.4**	0.001
	Lymphocytopenia < 20%	46.9	**68.7**	<0.001
	Creatine Kinase > 25 U/L	**11.8**	3.5	0.013
Helene Faury et al. [11]	White Blood cell count (G/L, median, [IQR])	5.88 [4.41–7.68]	**6.72 [5.15–9.42]**	0.01
	Neutrophil (G/L, median, [IQR])	4.11 [2.99–5.65]	**5.06 [3.43–7.25]**	0.02
	Platelets (G/L, median, [IQR])	179 [145–225]	**199 [168–239]**	0.04
	Sodium (U/L, median, [IQR]))	137 [135–139]	**138 [136–140]**	0.006
	Troponin (ng/L, median, [IQR])	9.2 [6.5–22.4]	**34.4 [8.8–72.2]**	0.007
	Albumin (g/L, median, [IQR])	30 [27–33]	**37 [33–39]**	0.04
	Aspartate aminotransferase (U/L, median, [IQR])	**45 [34–76]**	**34 [29–49]**	0.02
	LDH (U/L, median, [IQR])	**397 [305–544]**	**298 [248–383]**	0.04
Xiao Tang et al. [13]	PaO_2_/FiO_2_ (Median, mm Hg)	**198.5**	107	<0.001
	Aspartate transaminase (U/L)	25.5	**70**	<0.001
	LDH (U/L)	483	**767**	<0.001
	Troponin I (ng/mL)	0.03	**0.14**	<0.001
	CD3^+^ (Median, cells/mL)	193	**303**	0.007
	CD4^+^/CD3^+^ (Median, cells/mL)	97	**185**	<0.001
Natalie L. Cobb et al. [14]	White blood cells at admission (median, [IQR])	7240 [5430–11,820]	**9035 [6590–14,900]**	0.007
	Neutrophil at admission (median, [IQR])	5405 [3880–9580]	**7210 [4990–11,890]**	0.02
	Early sputum cultures (positive)	27.2	**72.2**	0.005
Raija Auvinen et al. [18]	Leukocytes count ×10^9^ /L (Median, [IQR])	5.1 (4.0–6.3)	**6.7 (5.4–10.9)**	0.002
	Leukocytosis	11	**39**	0.019
	Thrombocytopenia < 150 × 10^9^ /L	**39**	12	0.019
	Alanine aminotransferase (U/L, [IQR])	**42 [19–127]**	23 [12–123]	0.011
Zhilan Yin et al. [19]	Neutrophil (×10^9^ cells/L, median, [IQR])	3.57 [2.72–4.92]	**4.75 [3.15–7.00]**	0.037
	Procalcitonin (ng/mL, median, [IQR])	0.04 [0.03–0.09]	**0.11 [0.09–0.37]**	0.002
Yi-Hua Lin et al. [20]	White blood cells (×10^9^ cells/L, mean)	4.87 ± 2.04	**7.59 ± 5.12**	0.026
	Leukocytosis	3	**32**	0.002
	Neutrophil (×10^9^ /L, mean)	3.16 ± 1.73	**6.20 ± 4.84**	0.009
	Lymphocyte (×10^9^ /L, mean)	**1.19 ± 0.59**	0.88 ± 0.52	0.049
	Anemia	0	**41**	<0.001
	CRP (mg/L, median, [IQR]))	9.56 [3.82–22.42]	**55.3 [33.97–102.77]**	0.001
	Procalcitonin (ng/L, median, [IQR])	0.05 [0.05–0.06]	**0.25 [0.08–2.28]**	<0.001
	Urea Nitrogen (mmol/L, mean)	3.76 ± 1.37	**6.36 ± 3.30**	0.002
	LDH (U/L, median, [IQR]))	158.0 [142.0–196.0]	**243.5 [198.3–328.8]**	<0.001
	PaO_2_/FiO_2_ < 200 mm Hg	4	**22**	0.022
Jaehee Lee et al. [21]	White blood cell (Median, cells/uL)	7470	2680	0.027

Bold: illness (COVID-19 or influenza) with significant difference. Abbreviations: Leukocytosis: leukocyte count > 10 × 10^9^ /L; CRP: C-reactive protein; LDH: Lactate dehydrogenase; IQR: Interquartile range; MCV: Mean corpuscular volume; MCH: Mean corpuscular hemoglobin; MCHC: Mean corpuscular concentration; RDW-CV: Coefficient variation of red blood cell volume width; RDW_SD: Standard deviation in red cell distribution width; PDW: Platelet distribution width; GGT: γ-glutamyl transpeptidase; HDL-C: High-density lipoprotein cholesterol; LDL-C: Low-density lipoprotein cholesterol; HCO_3_: Bicarbonate ion; ESR: Erythrocyte sedimentation rate; APTT: Activated partial thromboplastic time.

**Table 4 viruses-13-00452-t004:** Significant radiological findings between COVID-19 and influenza adult groups.

References	Significant Radiological Findings	COVID-19 (%)	Influenza (%)	*p*-Value < 0.05
Jianguo Zhang et al. [10]	Rounded opacities	**37.9**	19.1	<0.001
	Bronchiolar wall thickening	**33.6**	13	<0.0001
	Air bronchogram	**29.9**	13	<0.001
	Consolidation	**26.1**	15.7	0.031
	Interlobular septal thickening	**24.2**	13.9	0.029
	Crazy paving pattern	**22.3**	9.6	0.004
	Tree-in-bud	**13.7**	5.2	0.018
	GGO with consolidation	25.6	**39.1**	0.011
Helene Faury et al. [11]	Pulmonary nodules	8.8	50.0	0.001
Mengqi Liu et al. [12]	Predominant distribution:			
	– Central	2	**6**	0.022
	– Peripheral	**45**	20	
	– Mixed	53	**74**	
	Interlobular septal thickening	**66**	43	0.014
	Rounded opacities	**35**	17	0.048
	Nodules	28	**71**	<0.001
	Tree-in-bud	9	**40**	<0.001
	Pleural effusion	6	**31**	<0.001
	Pure GGO without nodules	**29**	11	<0.001
	Pure GGO + interlobular septal thickening	**21**	6	0.042
	Rounded opacities without nodules	**22**	0	0.002
	Interlobular septal thickening without nodules	**45**	6	<0.001
	Rounded opacities + interlobular septal thickening + absence of pleural effusion	**19**	3	0.021
Xiao Tang et al. [13]	GGO	**94.5**	45.3	<0.001
	Consolidation	28.8	**45.3**	0.042
Natalie L Cobb et al. [14]	Bilateral opacities	90	52	<0.001
Hao Wang et al. [16]	Lesion Distribution:			
	– Central	7.7	**75**	0.000
	– Peripheral	**38.5**	3.3	
	– Diffuse	0	**21.7**	
	– Non-specific	**53.8**	0	
	Lobe predomination:			
	– Superior lobe	23.1	**23.9**	0.001
	– Inferior lobe	15.4	**57.6**	
	– Middle lobe	**7.7**	7.6	
	– Balanced predomination	**53.8**	10.9	
	Lesion margin:			
	– Clear	**46.2**	10.9	0.004
	– Vague	53.8	**89.1**	
	GGO Involvement pattern:			
	– Patchy	**38.5**	5.4	0.000
	– Cluster like	7.7	**77.2**	
	– GGO + consolidation opacities	**46.2**	6.5	
	– Whole consolidation	7.7	**10.9**	
	Lesion Contour:			
	– Shrinking	**69.2**	1.1	0.000
	– Non-shrinking	30.8	**98.9**	
	Bronchial wall thickening	0	**32.6**	0.018
Liaoyi Lin et al. [17]	Close to the pleura	**69**	40	0.005
	Mucoid impaction	2	**13**	0.047
	Pleural effusion	0	**22**	<0.001
	Axial distribution:			
	– Inner	6	**7**	<0.001
	– Outer	**67**	24	
	– Diffuse	12	**36**	
	– Random	15	**33**	
Raija Auvinen et al. [18]	Linear opacities	14	**42**	0.024
	GGO/Consolidation	**68**	21	< 0.001
Zhilan Yin et al. [19]	Vascular enlargement	67	**93**	0.037
	Pleural Thickening	63	**90**	0.03
	Linear opacification	50	**90**	0.002
	Crazy-paving sign	30	**60**	0.021
	Pleural effusion	**53**	13	0.002
	Bronchiectasis	**30**	3	0.012
Yi-Hua Lin et al. [20]	GGO	**71**	23	< 0.001
	Infiltration	29	**68**	0.003
	GGO + reticular pattern	**63**	0	< 0.001
	Interlobular septal thickening	**71**	27	0.001

Bold: illness (COVID-19 or influenza) with significant difference. Abbreviations: GGO: Ground-glass opacity.

## Data Availability

Data available on request due privacy to restrictions. The data presented in this case study are available on request from the corresponding author.

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
