# Peer review of "A Comparative Systematic Review of COVID-19 and Influenza"

_viruses, 2021, doi:10.3390/v13030452_

Round 1

Reviewer 1 Report

Osman et al. studied the “A Comparative Systematic Review of COVID-19 and Influenza”.

In this Review, the authors have attempted to compare the “clinical features and outcomes, laboratory and radiological findings of COVID-19 and influenza in adult patients”. The Review is timely captured, and it is needed by many of the researchers to probe in-depth in the flu research in terms of prophylactic and therapeutic perspective. Although the manuscript is an acceptable format, still it requires some minor edits.

Here are the some minor comments

The abstract is not in the journal format. Please format as per the requirement.

Line 15: “othorynaryngological symptoms” do you mean “Otorhinolaryngological symptoms”

Figure 1: On the printed page, the figure looks blurred. Please increase the resolution.

Table 1: Although many of the researchers known the terminology (Systematic review, Meta-analysis, Retrospective, Prospective, and others) used in the table, it would be helpful to define them in the legend for the young researchers.

Line 101: Only a few authors ----- Only a few studies

Line 111: Covid-19 ----- COVID-19

Line 159: BA and MO means basophils and monocytes

Line 240-241: This reference deserves a citation. https://doi.org/10.1016/j.xcrm.2020.100016

Author Response

To the Editor-in-Chief of the journal ‘Viruses’ and reviewers;

First and foremost, all authors thank you for the detailed interest you showed for our manuscript. Please find our answers hereinafter. Besides, we took into account your review and added precisions to the manuscript according to the remarks you resubmitted.

Response to Reviewer 1 Comments

Osman et al. studied the “A Comparative Systematic Review of COVID-19 and Influenza”.

In this Review, the authors have attempted to compare the “clinical features and outcomes, laboratory and radiological findings of COVID-19 and influenza in adult patients”. The Review is timely captured, and it is needed by many of the researchers to probe in-depth in the flu research in terms of prophylactic and therapeutic perspective. Although the manuscript is an acceptable format, still it requires some minor edits.

Here are the some minor comments:

Point 1: The abstract is not in the journal format. Please format as per the requirement.

Response 1: We have carefully read the guidelines outlined in the 'Instructions for
Authors' on the journal website and changed the abstract format with no more than 200 words.

Abstract: 1) Background: Both SARS-CoV-2 and influenza virus share similarities such as clinical features and outcome, laboratory and radiological findings. 2) Methods: Literature search was done using PubMed to find MEDLINE indexed articles relevant to this study. As of November 25, 2020, the search has been conducted by combining the MeSH words "COVID-19" and “Influenza”. 3) Results: Eighteen articles were finally selected in adult patients. Comorbidities such as cardiovascular diseases, diabetes and obesity were significantly higher in COVID-19 patients, while pulmonary diseases and immunocompromised conditions were significantly more common in influenza patients. The incidence rates of fever, vomiting, ocular and otorhinolaryngological symptoms were found to be significantly higher in influenza patients when compared with COVID-19 patients. However, neurologic symptoms and diarrhea were statistically more frequent in COVID-19 patients. Level of white cell count and procalcitonin were significantly higher in influenza patients, while thrombopenia and elevated transaminases were significantly more common in COVID-19 patients. Ground-grass opacities, interlobular septal thickening and a peripheral distribution were more common in COVID-19 patients than in influenza patients where consolidations and linear opacities were described instead. COVID-19 patients were significantly more often transferred to intensive care unit with a higher rate of mortality. 4) Conclusions: This study estimated differences of COVID-19 and influenza patients which can help clinicians during the co-circulation of the two viruses.

Point 2: Line 15: “othorynaryngological symptoms” do you mean “Otorhinolaryngological symptoms”

Response 2: Line 15 (Abstract, but also in discussion (line 162) and conclusion (line 240)): Yes, done (revised as requested). “othorynaryngological symptoms” has been removed and replaced with “Otorhinolaryngological symptoms”.

Point 3: Figure 1: On the printed page, the figure looks blurred. Please increase the resolution.

Response 3: As you recommended, we have changed the figure to increase the resolution.

Point 4: Table 1: Although many of the researchers known the terminology (Systematic review, Meta-analysis, Retrospective, Prospective, and others) used in the table, it would be helpful to define them in the legend for the young researchers.

Response 4: As you recommended, we have defined all the terminology used in Table 1. We have added this at the bottom of the table.

Cohort studies are used to investigate causes of disease and establish association between risk factors and health outcomes. An outcome-free study population is first identified by the exposure/event of interest and followed in time until the outcome of interest happens. They can be prospective (carried out form the present time into the future) or retrospective (carried out at the present time and look to the past to examine medical events or outcome). Case-Control studies first identify subjects by outcome status (cases), then select from the same source population, subjects without the outcome (control). Cross-sectional study or prevalence study is an observational study that collects data on the subjects of interest at a specific point in time.

Point 5: Line 101: Only a few authors ----- Only a few studies

Response 5: Line 101 (Results): Done (revised as requested). ‘Only a few authors’ has been removed and replaced with ‘Only a few studies’

Point 6: Line 111: Covid-19 ----- COVID-19

Response 6: Line 111 (Results): Done (revised as requested). ‘Covid-19’ has been removed and replaced with ‘COVID-19’

Point 7: Line 159: BA and MO means basophils and monocytes

Response 7: Yes. BA and MO stand for basophils and monocytes. We have defined these abbreviations in this sentence: ‘Two parameters (monocyte (MO) count and percentage of basophils (BA)) were combined to clarify the diagnostic efficacy, with a sensitivity of 71.6% and a specificity of 74.8%’.

We have also ensured that abbreviations are defined in parentheses the first time they appear in the abstract, main text, and in figure or table captions.

Point 8: Line 240-241: This reference deserves a citation. https://doi.org/10.1016/j.xcrm.2020.100016

Response 8: As you recommended, we have added this reference (Ref 34) to explain that SARS-COV-2 invasion activated T cell-mediated immunity, which resulted in increasing production of inflammatory cytokine.

Other Point: We also received an email from the editor which explains us that the second paragraph in the discussion section shows high similarity with you’re a recent publication of Zayet al.

Response: In this paragraph, we try to explain that differences in the clinical presentation in SARS-CoV-2 and influenza infections can be explained by the pathophysiological distribution of the entry receptors.

In order to do this, we have changed the whole paragraph as well (Line 166):

Indeed, the differences in the clinical presentation of these two viral infections can be explained, in large part, by the pathophysiological distribution of the entry receptors for each virus. Human influenza A virus binds to cell receptors alpha2,6-linked via sialic acid linked glycoproteins. The distribution of sialic acid on cell surfaces is one determinant of host tropism and understanding its expression on human cells and tissues is important for understanding influenza pathogenesis. These receptors were especially expressed on the respiratory tract, from the nasopharynx, trachea to the bronchi, except the alveoli (α2,3-linked sialic acid receptors predominant on alveolar cells) (22). The short incubation period of influenza infection with the predominant respiratory manifestations (sore throat, sneezing, sputum production, rhonchi on pulmonary auscultation) is well explained by this distribution. On the other hand, Angiotensin-converting enzyme 2 (ACE2) protein, known as the key regulator enzyme of the renin–angiotensin–aldosterone system (RAAS) is the functional receptor of SARS-CoV-2 and its expression and activity will mediate directly the SARS‐CoV‐2 infection. Regarding the tissue distribution of ACE2 protein, ACE2 is highly expressed on lung alveolar epithelial cells, small intestinal epithelial cells and endothelial cells (including in the central nervous system) but poorly found on the surface of nasopharyngeal cells (23). When the SARS spike protein binds to the ACE-2 receptor, the complex is proteolytically processed by type 2 transmembrane protease TMPRSS2 leading to cleavage of ACE-2 and activation of the spike. This mechanism is also described in influenza physiopathology. All these findings trigger a longer incubation period of SARS-CoV-2 infection, and the observed symptoms in COVID-19 patients of dyspnea, dry cough, diarrhea, and bilateral crackling sound on pulmonary auscultation, but also neurologic symptoms such as new loss of smell and taste. ACE is also abundantly present in the basal layer of the non-keratinizing squamous epithelium of nasal and oral mucosa (24). Indeed, most reports have so far linked new loss of taste or smell to neurological symptoms instead of rhinolaryngological symptoms (25). In any case, in this epidemic context, patients presenting with dysgeusia and/or anosmia may be considered as patients infected with COVID-19, until microbiological confirmation has been obtained. In another study including 217 patients presenting influenza like illness, we demonstrated that the specificity of the combination of anosmia and dysgeusia reached 91% for a positive SARS-CoV-2 RT-PCR result (26). Unspecific symptoms such as fever and musculoskeletal symptoms or pain syndrome defined by fatigue, myalgia and/or arthralgia are associated with a cascade of inflammatory mediators and were not directly linked to the distribution of viral receptors. These clinical presentations can be equally described in the two illnesses.

I am looking forward to your response and honorable review. Please accept the assurances of my highest consideration. Thank you again for your contributions and interest.

Sincerely,

Souheil ZAYET, MD

Infectious Disease Department, Nord Franche-Comte Hospital, FRANCE

Reviewer 2 Report

Interesting publication summarizing what is known so far regarding the similarities and differences between SARS-CoV-2 and the influenza virus.

Here are the main critical remarks:
1. English should be checked by a native speaker,
2. please explain why only one bibliographic database was searched,
3. please explain why only publications with free access to the full text were used in the review,
4. systematic review has not been registered in PROSPERO - please explain,
5. Fig. 1 is illegible,
6. It is not clear what was the purpose of including other review articles or meta-analysis in the review,
7. the authors did not provide a bias analysis, e.g. using NOS,
8. have the authors analyzed other literature sources, e.g. a bibliography of the included manuscripts,
9. the demographic analysis lacks basic data about patients - age, gender, ethnicity, etc.,

10. In my opinion, the  ables require redrafting - it is worth presenting the parameters in such a way that they can also be compared between individual studies,
11. the paper limits are not described carefully and need to be supplemented,
12. there is no summary table in the paper in accordance with PRISMA rules - therefore the question is whether all the conditions for such a review have been met?
13. Literature is very limited. I am attaching manuscripts worth quoting:

doi: 10.3390/pathogens9060493,

doi: 10.1055/a-1366-9656,

 doi: 10.3390/pathogens9030231.

Author Response

To the Editor-in-Chief of the journal ‘Viruses’ and reviewers;

First and foremost, all authors thank you for the detailed interest you showed for our manuscript. Please find our answers hereinafter. Besides, we took into account your review and added precisions to the manuscript according to the remarks you resubmitted.

Response to Reviewer 2 Comments

Interesting publication summarizing what is known so far regarding the similarities and differences between SARS-CoV-2 and the influenza virus.

Here are the main critical remarks:

Point 1. English should be checked by a native speaker.

Response 1:

As you recommended, we have consulted with a colleague fluent in English to assist in drafting this revision. We have also ensured that abbreviations are defined in parentheses the first time they appear in the abstract, main text, and in figure or table captions. With this revised manuscript, we have respected all the authors’ instructions. We hope that the text is more consistent and understandable.

Point 2. Please explain why only one bibliographic database was searched,

Response 2: We consider PubMed as the main reference for medical bibliographic research.

This has been emphasized in the article's limitations.

Point 3. Please explain why only publications with free access to the full text were used in the review,

Response 3: The literature search was conducted by Dr Osman Molka in Tunisia, where only free articles were accessible. The choice to only include free accessible articles was to ensure legal research. We have chosen to mention it in the methods in order to be 100% transparent.

However, in fact, due to the exceptional pandemic context most of COVID-19 articles are with free access.

This has been emphasized in the article's limitations.

Point 4. Systematic review has not been registered in PROSPERO - please explain,

Response 4: We did not consider it when we wrote the manuscript.

Point 5. Fig. 1 is illegible,

Response 5: As you recommended, we have changed the figure to increase the resolution.

Point 6. It is not clear what was the purpose of including other review articles or meta-analysis in the review,

Response 6: When we did our bibliographic research, the number of articles concerning our main subject which was COVID-19 versus Influenza was very limited. We then chose to include all types of articles that corresponded to our objective and our inclusion criteria’s.

Of course review and meta-analysis that were included were not duplicated from the other articles included.

Point 7. The authors did not provide a bias analysis, e.g. using NOS,

Response 7: We did not do meta-analyses, therefore we didn’t extract a database nor did we do an analysis. We just summarized what we found and focused on what was statistically significant.

This has been emphasized in the article's limitations.

Point 8. Have the authors analyzed other literature sources, e.g. a bibliography of the included manuscripts,

Response 8: No, we only analysed our selected articles and quoted them when needed, in accordance with the ethics.

Point 9. The demographic analysis lacks basic data about patients - age, gender, ethnicity, etc.,

Response 9: We have focused on what was statistically significant and mentioned the demographics when available and when a difference between COVID-19 group and influenza group was significant

Concerning the age: we have mentioned in results that Qu et al. reported that while the incidences of COVID-19 and influenza were comparable among the 18-65 and >65 year age groups, the incidences of influenza were much higher than COVID-19 among those aged under 18 years old. However, Lee et al, reported that the median age was significantly higher in patients with COVID-19 compared to patients with influenza (68 [IQR: 59-75] years vs 57 [IQR: 44-63] years, p=0.016)’

We have also added this sentence in the results section (line X, next ref 21): ‘No significant difference was found concerning other demographic characteristics such as gender and ethnicity in our review.’

Point 10. In my opinion, the Tables require redrafting - it is worth presenting the parameters in such a way that they can also be compared between individual studies,

Response 10: In our first submission, we separated our results in three different tables (Table 2. Significant demographic and baseline characteristics, clinical features and outcome, then Table 3. Significant laboratory findings and finally Table 4. Significant radiological findings).

In this revision, we have chosen to draft the tables this way (on one table), as you recommended.

I suggest that you choose the most appropriate presentation.

As you recommended, here is a suggestion of our summary table:

Table 2. Significant demographic and baseline characteristics, clinical features and outcome in COVID-19 and influenza adults groups.

Reference

Significant clinical features/Outcome, laboratory and radiological findings

COVID-19 (%)

Influenza (%)

p-value <0.05

Jordan Cates et al. (6)

Admitted to ICU

Hospital mortality

Duration of hospitalization (median days,[IQR])

36.5

21.0

8.6 [3.9-18.6]

17.6

3.8

3.0 [1.8-6.5]

<0.001

<0.001

<0.001

Ying Luo et al. (7)

White blood cell count (x 109/L, median, [IQR])

Neutrophil (x 109/L, median, [IQR]))

Lymphocyte (%, median, [IQR])

Monocyte (x 109/L, median, [IQR])

Eosinophil (x 109/L, median, [IQR])

Eosinophil (%, median, [IQR])

Basophil (%, median, [IQR])

Red blood cell count (x 1012/L, median, [IQR])

Hemoglobin (g/L, median, [IQR])

Hematocrit (%, median, [IQR])

MCV (fL, median, [IQR])

MCH (pg, median, [IQR])

MCHC (g/L, median, [IQR])

RDW-CV (Median, [IQR])

RDW-SD (fL, median, [IQR])

PDW (fL, median, [IQR])

Alanine aminotransferase (U/L, median, [IQR])

Aspartate aminotransferase (U/L, median, [IQR])

Total Protein (g/L, mean)

Globulin (g/L, median, [IQR])

Indirect Bilirubin (mmol/L, median, [IQR])

GGT (U/L, median, [IQR])

Alkaline Phosphatase (U/L, median, [IQR])

LDH (U/L, median, [IQR])

Triglyceride (mmol/L, median, [IQR])

HDL-C (mmol/L, median, [IQR])

LDL-C (mmol/L, median, [IQR])

Creatinine (mmol/L, median, [IQR])

Urea (mmol/L, median, [IQR])

Uric acid (mmol/L, median, [IQR])

Calcium (mmol/L, median, [IQR])

Magnesium (mmol/L, median, [IQR])

Chlorine (mmol/L, median, [IQR])

Potassium (mmol/L, median, [IQR])

Sodium (mmol/L, median, [IQR])

Phosphate (mmol/L, median, [IQR])

HCO3 (mmol/L, median, [IQR])

Hypersensitive CRP (mg/L, median, [IQR])

ESR (mm/h, median, [IQR])

Prothrombin time (s, mean)

APTT (s, mean)

Thrombin time (s, mean)

Prothrombin activity (%, mean)

Fibrinogen (g/L, mean)

D-Dimer (mg/L, median, [IQR])

5.45 [4.46-7.17]

3.68 [2.68-5.16]

22.0 [14.6-29.4]

0.47 [0.34-0.61]

0.01 [0.00-0.05]

0.2 [0.0-2.9]
0.2 [0.0-0.3]

4.43 [4.00-4.84]

134 [122-146]

39.7 [36.2-43.1]

89.1 [86.4-91.7]

30.6 [29.5-31.6]

343 [335-351]

12.2 [11.9-12.8]

39.5 [37.8-41.8]

12.0 [10.8-13.6]

25 [18-38]

27 [21-36]

69.3 ± 5.6

32.4 ± 4.4

5.5 [4.2-7.3]

30 [21-48]

65 [56-78]

260 [217-327]

1.75 ± 088

0.99 ± 0.19

2.45 ± 0.55

72 [61-87]

5.89 ± 3.84

253 [206-313]

2.14 ± 0.11

0.87 ± 0.07

100.4 ± 4.2

4.21 ± 0.42

139.7 ±3.9

1.04 ± 0.26

24.5 ± 2.9

20.0 [5.8-45.8]

35 [24-47]

14.06 ± 1.09

39.9 ± 4.5

16.9 ± 1.4

91 ± 11

4.71 ± 1.08

1.24 [0.65-2.75]

6.14 [4.66-8.24]

4.09 [2.85-6.11]

20.5 [13.3-28.6]

0.52 [0.37-0.69]

0.02 [0.00-0.07]

0.3 [0.0-1.2]

0.2 [0.1-0.3]

4.37 [3.96-4.78]

131 [119-143]

39.1 [35.5-42.4]

89.6 [86.7-92.4]

30.4 [29.3-31.3]

337 [329-346]

12.5 [12.0-13.2]

40.9 [38.8-43.2]

12.3 [11.0-13.9

24 [16-36]

25 [19-35]

68.5 ± 6.4

31.8 ± 4.8

4.9 [3.8-6.9]

35 [21-54]

75 [63-96]

235 [196-298]

1.63 ± 0.84

0.97 ± 0.22

2.41 ± 0.68

69 [59-82]

5.54 ± 3.41

260 [219-304]

2.17 ± 0.11

0.86 ± 0.09

101.4 ± 3.7

4.15 ± 0.40

139.1 ± 3.4

1.05 ± 0.20

24.0 ± 3.1

15.7 [4.8-40.1]

27 [17-40]

14.09 ± 1.83

39.6 ± 5.0

16.6   ± 2.0

92 ± 14

4.27 ± 1.18

1.72 [0.85-3.30]

<0.001

<0.001

0.009

<0.001

<0.001

<0.001

<0.001

0.012

<0.001

0.002

0.003

0.002

<0.001

<0.001

<0.001

0.021

0.019

<0.001

0.003

<0.001

<0.001

0.003

<0.001

<0.001

<0.001

0.002

0.004

<0.001

0.001

0.031

<0.001

0.001

<0.001

<0.001

<0.001

0.002

<0.001

0.024

<0.001

<0.001

0.020

<0.001

<0.001

<0.001

<0.001

Jiangnan Chen et al. (8)

Monocyte (x 109/L, median, [IQR])

Monocyte (%, median, [IQR])

Neutrophil (x 109/L, median, [IQR])

Neutrophil (%, mean)

Lymphocyte (%, mean)

Eosinophil (%, median, [IQR])

Basophil (%, median, [IQR])

0.36 [0.28-0.48]

7.60 [6.20-9.95]

2.93 [2.26-3.79]

64.50 ± 11.64

26.30 ± 10.52

0.60 [0.30-1.15]

0.20 [0.10-0.30]

0.55 [0.4-0.71]

9.0 [7.20-11.40]

4.26 [3.00-5.74]

68.42 ± 14.69

21.07 ± 12.85

0.40 [0.10-1.10]

0.10 [0.10-0.30]

0.000

0.000

0.000

0.011

0.000

0.038

0.001

Jiajia Qu et al. (9)

Fever

Persistent fever

Elevated lymphocyte

Abnormal Urinary test

Urine protein positive

Elevated procalcitonin

Elevated white blood cells

78.5

50.4

0

32.11

16.26

40.83

75.00

89.2

74.2

5.00

21.67

8.33

10.98

26.83

<0.05

<0.01

<0.01

<0.05

<0.05

<0.01

<0.01

Jianguo Zhang et al. (10)

Cough

Expectoration

Dyspnea

Chest pain

Vomiting

Leukocytosis > 9.5 x 109/L

Neutrophilia > 75%

Lymphocytopenia < 20%

Creatine Kinase > 25 U/L

Rounded opacities

Bronchiolar wall thickening

Air bronchogram

Consolidation

Interlobular septal thickening

Crazy paving pattern

Tree-in-bud

GGO with consolidation

69.7

22.7

14.7

13.7

1.4

16.1

32.2

46.9

11.8

37.9

33.6

29.9

26.1

24.2

22.3

13.7

25.6

86.1

74.8

27.8

27.8

9.6

30.4

50.4

68.7

3.5

19.1

13.0

13.0

15.7

13.9

9.6

5.2

39.1

0.001

<0.001

0.004

0.002

<0.001

0.003

0.001

<0.001

0.013

<0.001

<0.0001

<0.001

0.031

0.029

0.004

0.018

0.011

Hélène Faury et al. (11)

Chronic pulmonary diseases

Overweight/Obesity

Median BMI

Fatigue

Diarrhea

Faintness

Anosmia/Ageusia

Sputum production

Nasal Congestion

Secondary respiratory failure

Acute Kidney failure

Pulmonary embolism

Heart congestion

Admitted to ICU

Duration of hospitalization (days,[IQR])

Oxygen therapy

Mortality rate

White Blood cell count (G/L, median, [IQR])

Neutrophil (G/L, median, [IQR])

Platelets (G/L, median, [IQR])

Sodium (U/L, median, [IQR]))

Troponin (ng/L, median, [IQR])

Albumin (g/L, median, [IQR])

Aspartate aminotransferase (U/L, median, [IQR])

LDH (U/L, median, [IQR])

Pulmonary nodules

12.0

40.8

27.3

63.6

25.8

12.1

7.0

12.0

8.3

21.0

17.0

6.0

2.0

31.0

10 [4-17]

65.0

20.0

5.88 [4.41-7.68]

4.11 [2.99-5.65]

179 [145-225]

137 [135-139]

9.2 [6.5-22.4]

30 [27-33]

45 [34-76]

397 [305-544]

8.8

27.0

25.0

24.8

39.0

13.0

3.0

0

36.0

21

0

7.0

0

14.0

12.0

4 [1-11]

42.3

5.0

6.72 [5.15-9.42]

5.06 [3.43-7.25]

199 [168-239]

138 [136-140]

34.4 [8.8-72.2]

37 [33-39]

34 [29-49]

298 [248-383]

50.0

0.01

0.02

0.04

0.0006

0.03

0.02

0.01

0.0001

0.02

<0.0001

0.048

0.03

0.003

0.002

<0.0001

0.002

0.002

0.01

0.02

0.04

0.006

0.007

0.04

0.02

0.04

0.001

Pengfei Li et al. (5)

Cardiovascular disease/Hypertension

Diabetes

Asthma

Chronic Obstructive Pulmonary disease

Immunocompromised conditions

Fever

Cough

Shortness of breath

Rhinorrhea

Sore throat

Myalgia/Muscle pain

Vomiting

28.76

16.38

8.42

4.93

4.39

72.08

57.99

32.89

8.48

9.48

18.97

8.67

14.11
11.12

16.09

9.52
9.99

89.99

85.31

49.19

38.57

37.28

30.12

24.27

<0.0001

0.012

0.0033

0.0003

<0.0001

<0.0001

<0.0001

0.0249

<0.0001

<0.0001

0.0242

<0.0001

Mengqi Liu et al. (12)

Stuffy and runny nose

Predominant distribution:

-         Central

-         Peripheral

-         Mixed

Interlobular septal thickening

Rounded opacities

Nodules

Tree-in-bud

Pleural effusion

Pure GGO without nodules

Pure GGO + interlobular septal thickening

Rounded opacities without nodules

Interlobular septal thickening without nodules

Rounded opacities + interlobular septal thickening + absence of pleural effusion

7

2

45

53

66

35

28

9

6

29

21

22

45

19

23

6

20

74

43

17

71

40

31

11

6

0

6

3

0.002

0.022

0.014

0.048

<0.001

<0.001

<0.001

<0.001

0.042

0.002

<0.001

0.021

Xiao Tang et al. (13)

Productive cough

Fatigue

GI symptoms

Myalgia

PaO2/FiO2 (Median, mm Hg)

Aspartate transaminase (U/L)

LDH (U/L)

Troponin I (ng/mL)

CD3+ (Median, cells/mL)

CD4+/CD3+ (Median, cells/mL)

GGO

Consolidation

53.4

63.0

37.0

34.2

198.5

25.5

483

0.03

193

97

94.5

28.8

78.7

18.7

6.7

14.7

107.0

70.0

767

0.14

303

185

45.3

45.3

0.002

<0.001

<0.001

0.007

<0.001

<0.001

<0.001

<0.001

0.007

<0.001

<0.001

0.042

Natalie L. Cobb et al. (14)

ARDS*

Hospital mortality

White blood cells at admission (median, [IQR])

Neutrophil at admission (median ,[IQR])

Early sputum cultures (positive)

Bilateral opacities

63

40

7240 [5430-11820]

5405 [3880-9580]

27.2

90

26

19

9035 [6590-14900]

7210 [4990-11890]

72.2

52

<0.001

0.006

0.007

0.02

0.005

<0.001

Souheil Zayet et al. (15)

Frontal headache

Retro-orbital or temporal headache

Fever

Anosmia

Dysgeusia

Diarrhea

Sputum Production

Sneezing

Dyspnea

Sore throat

Conjunctival hyperemia

Tearing

Vomiting

Crackling sound

Ronchi sounds

25.7

18.6

75.7

52.9

48.6

40.0

28.6

18.6

34.3

20.0

4.3

5.7

2.8

38.6

1.4

9.3

3.7

92.6

16.7

20.4

20.4

51.9

46.3

59.3

44.4

29.6

24.1

22.2

20.4

16.7

0.021

0.013

0.042

<0.001

0.001

0.021

0.010

0.001

0.007

0.006

<0.001

0.004

0.001

0.032

0.002

Hao Wang et al. (16)

Cough

Lesion Distribution:

-         Central

-         Peripheral

-         Diffuse

-         Non-specific

Lobe predomination:

-         Superior lobe

-         Inferior lobe

-         Middle lobe

-         Balanced predomination

Lesion margin:

-         Clear

-         Vague

GGO Involvement pattern:

-         Patchy

-         Cluster like

-         GGO + consolidation opacities

-         Whole consolidation

Lesion Contour:

-         Shrinking

-         Non-shrinking

Bronchial wall thickening

30.8

7.7

38.5

0.0

53.8

23.1

15.4

7.7

53.8

46.2

53.8

38.5

7.7

46.2

7.7

69.2

30.8

0

82.6

75.0

3.3

21.7

0.0

23.9

57.6

7.6

10.9

10.9

89.1

5.4

77.2

6.5

10.9

1.1

98.9

32.6

0.000

0.000

0.001

0.004

0.000

0.000

0.018

Liaoyi Lin et al. (17)

Close to the pleura

Mucoid impaction

Pleural effusion

Axial distribution:

-         Inner

-         Outer

-         Diffuse

Random

69

2

0

6

67

12

15

40

13

22

7

24

36

33

0.005

0.047

<0.001

<0.001

Raija Auvinen et al. (18)

Pulmonary Diseases

Current smoking

Headache

ARDS

ICU admission

Duration of hospitalization (days,[IQR])

Leukocytes count x 109/L (Median, [IQR])

Leukocytosis

Thrombocytopenia < 150 x109/L

Alanine aminotransferase (U/L, [IQR])

Linear opacities

GGO/Consolidation

18

4

85

93

29

6 [4-21]

5.1 (4.0-6.3)

11

39

42 [19-127]

14

68

45

30

52

58

6

3 [2-4]

6.7 (5.4-10.9)

39

12

23 [12-123]

42

21

0.03

0.008

0.004

0.003

0.034

<0.001

0.002

0.019

0.019

0.011

0.024

<0.001

Zhilan Yin et al. (19)

Cough

Expectoration

Neutrophil (x109 cells/L, median, [IQR])

Procalcitonin (ng/mL, median, [IQR])

Vascular enlargement

Pleural Thickening

Linear opacification

Crazy-paving sign

Pleural effusion

Bronchiectasis

73.3

43.3

3.57 [2.72-4.92]

0.04 [0.03-0.09]

67

63

50

30

53

30

96.7

80.0

4.75 [3.15-7.00]

0.11 [0.09-0.37]

93

90

90

60

13

3

0.026

0.007

0.037

0.002

0.037

0.030

0.002

0.021

0.002

0.012

Yi-Hua Lin et al. (20)

Fever 38.0°C-38.9°C

Fever ≥ 39.0°C

Cough

Expectoration

Dyspnea

Chills

White blood cells (x 109 cells/L, mean)

Leukocytosis

Neutrophil (x 109/L, mean)

Lymphocyte (x 109/L, mean)

Anemia

CRP (mg/L, median, [IQR]))

Procalcitonin (ng/L, median, [IQR])

Urea Nitrogen (mmol/L, mean)

LDH (U/L, median, [IQR]))

PaO2/FiO2<200 mm Hg

GGO

Infiltration

GGO + reticular pattern

Interlobular septal thickening

43

11

51

28

9

23

4.87 ± 2.04

3

3.16 ± 1.73

1.19 ± 0.59

0

9.56 [3.82-22.42]

0.05 [0.05-0.06]

3.76 ± 1.37

158.0 [142.0-196.0]

4

71

29

63

71

32

45

100
91

59

55

7.59 ± 5.12

32

6.20 ± 4.84

0.88 ± 0.52

41

55.3 [33.97-102.77]

0.25 [0.08-2.28]

6.36 ± 3.30

243.5 [198.3-328.8]

22

23

68

0

27

0.014

0.014

<0.001

<0.001

<0.001

0.015

0.026

0.002

0.009

0.049

<0.001

0.001

<0.001

0.002

<0.001

0.022

<0.001

0.003

<0.001

0.001

Jaehee Lee et al. (21)

Median heart rate (bpm)

White blood cell (Median, cells/uL)

83

7470

107

2680

0.017

0.027

Bold: illness (COVID-19 or influenza) with significant difference. Abbreviations: ARDS: Acute respiratory distress syndrome; ICU: Intensive care unit; IQR: interquartile range; Leukocytosis: leukocyte count > 10 x 109/L; CRP: C-reactive protein; LDH: Lactate dehydrogenase; IQR: Interquartile range; MCV: Mean corpuscular volume; MCH: Mean corpuscular hemoglobin; MCHC: Mean corpuscular concentration; RDW-CV: Coefficient variation of red blood cell volume width; RDW_SD: Standard deviation in red cell distribution width; PDW: Platelet distribution width; GGT: g-glutamyl transpeptidase; HDL-C: High-density lipoprotein cholesterol; LDL-C: Low-density lipoprotein cholesterol; HCO3: Bicarbonate ion; ESR: Erythrocyte sedimentation rate; APTT: Activated partial thromboplastic time; GGO: Ground-glass opacity.

Point 11. The paper limits are not described carefully and need to be supplemented,

Response 11: We added this paragraph: ‘We only considered PubMed a reference for our bibliographic research and database with accessible free access articles. We also just summarized what we found in recent medical literature and focused on what was statistically significant. Finally, a meta-analysis is recommended to further define the differences and the degree between COVID-19 and influenza.’

Point 12. There is no summary table in the paper in accordance with PRISMA rules - therefore the question is whether all the conditions for such a review have been met?

Response 12: We did our manuscript in accordance with PRISMA rules. The figure 1 is a PRISMA flow chart and the Table 1 is a general summary of included citations in this review.

We used the PRISMA guidelines for systematic reviews. We did not use any database or extract data or did a data analysis. We did a synthesis of what was statistically significant between COVID-19 and influenza.

Point 13. Literature is very limited. I am attaching manuscripts worth quoting:

doi: 10.3390/pathogens9060493,

doi: 10.1055/a-1366-9656,

doi: 10.3390/pathogens9030231.

Response 13:

As you recommended, we have added all these references and the corresponding paragraphs:

Ref 29, Line 200

In another short review of COVID-19 haematological manifestations, Slomka et al. showed that the majority of patients (especially those presenting with an acute respiratory distress syndrome related to SARS-CoV-2) are likely to develop lymphocytopenia. The decreased number of lymphocytes is described in others diseases caused by other coronaviruses primarily infecting the human respiratory tract. This lymphopenia is probably explained by direct infection of lymphocytes and destruction of lymphoid organs caused by the “cytokine storm” and a major release of pro-inflammatory cytokines.

Ref 35, Line 213

This severe inflammatory state secondary to COVID-19 leads to a severe derangement of hemostasis that has been recently described as a state of hypercoagulopathy, defined as increased degradation products such as D-dimer and fibrinogen. In a large American cohort study of 9407 adult COVID-19 patients, the overall in-hospital venous thromboembolism (VTE) was approximately 3%. The authors concluded that key predictors of VTE or mortality included advanced age, increasing CCI, past history of cardiovascular disease, ICU level of care, and elevated level of D-dimer.

Ref 24, in the second paragraph (Discussion section)

When the SARS spike protein binds to the ACE-2 receptor, the complex is proteolytically processed by type 2 transmembrane protease TMPRSS2 leading to cleavage of ACE-2 and activation of the spike. This mechanism is also described in influenza physiopathology.

I am looking forward to your response and honorable review. Please accept the assurances of my highest consideration. Thank you again for your contributions and interest.

Sincerely,

Souheil ZAYET, MD

Infectious Disease Department, Nord Franche-Comte Hospital, FRANCE

Round 2

Reviewer 2 Report

The authors have addressed all the comments of the reviewer and revised the manuscript accordingly.